# Towards Unsupervised Model Selection for Domain Adaptive Object Detection

**Hengfu Yu**[*]     **Jinhong Deng**[*]     **Wen Li**[†]     **Lixin Duan**

University of Electronic Science and Technology of China

hfyu@std.uestc.edu.cn, {jhdengvision, liwenbnu, lxduan}@gmail.com

## Abstract

Evaluating the performance of deep models in new scenarios has drawn increasing attention in recent years due to the wide application of deep learning techniques in various fields. However, while it is possible to collect data from new scenarios, the annotations are not always available. Existing Domain Adaptive Object Detection (DAOD) works usually report their performance by selecting the best model on the validation set or even the test set of the target domain, which is highly impractical in real-world applications. In this paper, we propose a novel unsupervised model selection approach for domain adaptive object detection, which is able to select almost the optimal model for the target domain without using any target labels. Our approach is based on the flat minima principle, *i.e.*, models located in the flat minima region in the parameter space usually exhibit excellent generalization ability. However, traditional methods require labeled data to evaluate how well a model is located in the flat minima region, which is unrealistic for the DAOD task. Therefore, we design a Detection Adaptation Score (DAS) approach to approximately measure the flat minima without using target labels. We show via a generalization bound that the flatness can be deemed as model variance, while the minima depend on the domain distribution distance for the DAOD task. Accordingly, we propose a Flatness Index Score (FIS) to assess the flatness by measuring the classification and localization fluctuation before and after perturbations of model parameters and a Prototypical Distance Ratio (PDR) score to seek the minima by measuring the transferability and discriminability of the models. In this way, the proposed DAS approach can effectively represent the degree of flat minima and evaluate the model generalization ability on the target domain. We have conducted extensive experiments on various DAOD benchmarks and approaches, and the experimental results show that the proposed DAS correlates well with the performance of DAOD models and can be used as an effective tool for model selection after training. The code will be released at `https://github.com/HenryYu23/DAS`.

## 1 Introduction

With the explosion of deep neural networks [27, 56, 18], object detection [24, 47, 3, 54, 5] has achieved promising results and shown great potential in many downstream tasks such as autonomous driving [2, 53], video understanding [55, 6], robot navigation [42, 41], *etc*. However, the well-trained object detection models frequently face previously unseen domains in real-world scenarios and often suffer from dramatic performance degradation when being deployed in a novel domain [11]. This is because the training set (*i.e.*, source domain) and the test set (*i.e.*, target domain) have distinct domain distributions. To address this problem, Domain Adaptive Object Detection (DAOD) [11, 50, 14, 38, 4]

---

[*]Equal contribution.
[†]The corresponding author.

38th Conference on Neural Information Processing Systems (NeurIPS 2024).

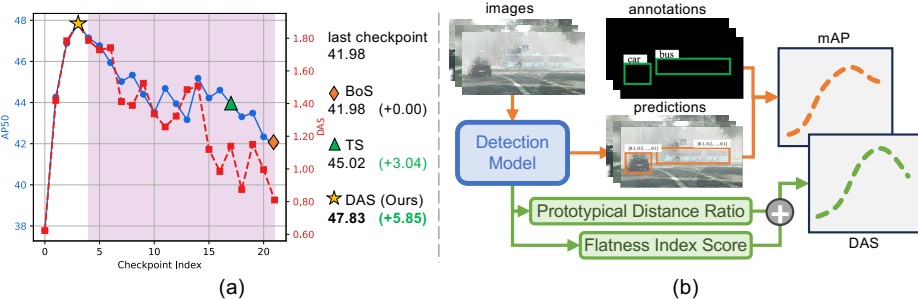

Figure 1: (a) The performance of classic DAOD method AT [38] on Real-to-Art (P2C) adaptation task during training. It suffers from performance degradation as the training goes on. The proposed DAS outperforms previous unsupervised model evaluation methods and selects desirable checkpoints without accessing any labels in the target domain. (b) The motivation of the work. We propose a Detection Adaptation Score including a Prototypical Distance Ratio (PDR) score and Flatness Index Score (FIS) to evaluate the model performance in an unsupervised way. It can be a good substitute metric for using annotations for DAOD model evaluation.

has been proposed to transfer the knowledge from the labeled source domain to an unlabeled target domain by leveraging adversarial training or pseudo-labeled approaches.

Although effective, these DAOD methods [11, 50, 14, 38, 4] evaluate the detector performance and conduct model selection relying on the labeled target data, which is usually unavailable and impractical for real-world domain adaptation scenarios. Due to the natural complexity of object detection, DAOD methods that leverage adversarial training and self-training techniques are often unstable and prone to overfitting to the target domains. As shown in Fig. 1 (a), DAOD methods usually suffer from a performance drop during training (marked as purple in Fig. 1 (a)). These issues limit the application of the DAOD model in real-world scenarios. Therefore, it is urgent and desirable to develop an unsupervised model selection method for DAOD, as shown in Fig. 1 (b).

Regarding the unsupervised model selection, several seminal works [22, 59, 60] attempt to evaluate the models from different aspects. For example, TS [59] examines the spatial uniformity of the unsupervised domain adaptive classifier, as well as the transferability and discriminability of deep representation. ATC [22] learns a threshold on the confidence of the model and predicts accuracy as the fraction of unlabeled examples for which model confidence exceeds that threshold. In object detection, however, the evaluation involves not only classification but also precise localization of objects within an image. This crucial distinction renders these methods ineffective in fully assessing an object detection model. A recent work [60] proposes a BoS metric to evaluate the generalization of the detection model by measuring the stability of the box under feature dropout. However, it does not consider the domain discrepancy between the source and the target domain, making the metric unreliable for DAOD model selection (see Fig. 1 (a) and our experiments in Sec. 4.2).

In this paper, we propose a novel Detection Adaptation Score (DAS) to evaluate the DAOD models without accessing any target labels, enabling select almost the optimal model for the target domain in an unsupervised way. The proposed DAS is based on the flat minima principles, *i.e.*, models that are located in the flat minima region in the parameter space exhibit better generalization ability than that in sharp minima region [21, 7]. However, the traditional flat minima search method requires labeled data to evaluate how well a model is located in the flat minima region, which is unrealistic for DAOD tasks. Therefore, we investigate how to measure the flat minima approximately without using target labels. We derive a generalization error bound that shows the flatness can be deemed as model variance, while the minima depend on the domain distribution distance for the DAOD tasks. Therefore, we first propose a Flatness Index Score (FIS) to assess the flatness by observing the classification and localization fluctuation before and after perturbations of model parameters. Then, we propose a Prototypical Distance Ratio (PDR) score to seek the minima models by measuring the transferability and discriminability of the models in the target domain. To this end, the proposed DAS can effectively represent the degree of flat minima for DAOD models and evaluate the model performance on the target domain. To evaluate the effectiveness of the proposed DAS, we conduct extensive experiments on public DAOD benchmarks, including weather adaptation, real-to-art, and synthetic-to-real adaptation. The experimental results show that the proposed metric can effectively evaluate the performance of DAOD models without annotating the target domain.

The contributions of our work can be summarized as follows:

- With the pressing need for the application of DAOD in real-world scenarios, to our best knowledge, we are the first to evaluate the DAOD models without using target labels.

- We propose a novel Detection Adaptation Score (DAS) by seeking the flat minima without using any target labels to evaluate the performance of DAOD models on the target domain. A Flatness Index Score (FIS) and a Prototypical Distance Ratio (PDR) score are proposed to meet the requirements of flatness and minima, respectively.

- We have conducted extensive experiments on several DAOD benchmarks and approaches. Our DAS benefits from selecting the optimal checkpoints of model parameters to avoid negative transfer or pseudo-label error accumulation. The experimental results validate the effectiveness of the proposed DAS.

## 2 Related Work

**Object Detection.** Object detection [23, 47, 54, 46, 39] is a fundamental task in computer vision that involves identifying and locating multiple objects within an image or video. The object detection approaches can be roughly divided into two categories: one-stage and two-stage. The one-stage methods [54, 46, 39, 19] directly estimate the categories and the location of the objects, such as FCOS [54], CenterNet [19], and YOLO series [46]. The two-stage methods [23, 47, 3] first generate region proposals for objects and then classify the category and regress the bounding box coordinates based on the proposal features, *e.g.*, Faster RCNN [47] and Cascade RCNN [3]. Recently, an end-to-end object detection model based on Transformer (*i.e.*, DEtection TRansformer, DETR) [5] has been proposed to eliminate the complex anchor generation and post-processing operations such as non-maximum suppression (NMS). Many successive works [67, 63] further improve the training efficiency and accuracy of DETR. With the strong representation of deep neural networks, the object detection model has achieved promising results in many object detection benchmarks. However, these models usually suffer from performance degradation [11] because of the domain discrepancy between the training and testing domains.

**Domain Adaptive Object Detection.** Domain Adaptive Object Detection (DAOD) [11, 50, 4, 38, 15, 8, 66, 16, 30, 65] aims to transfer knowledge from the labeled source domain to an unlabeled target domain. Previous works can be roughly categorized into two aspects: domain alignment and self-training. The domain alignment [11, 50, 8, 66] directly minimizes the domain discrepancy between the source and target domains. They minimize the feature distribution mismatch via adversarial training [11, 50, 66, 16], prototype alignment [58], graph matching [35, 36, 37], *etc*. The self-training approaches [14, 38, 4, 15, 10] follow a Mean Teacher (MT) framework and generate pseudo-labels from the teacher network to supervise the training of the student network. UMT [14] leverages the style transfer algorithm to eliminate the MT model bias towards the source domain. Adaptive Teacher (AT) [38] combines adversarial training and self-training to improve the accuracy on the target domain. CMT [4] introduces contrastive learning to improve the instance feature representation. HT [15] reveals that the consistency between classification and localization is crucial for pseudo-label generation and proposes a reweight strategy based on the harmony measure between the classification and localization. While effective, these approaches [11, 50, 4, 38, 15, 16, 35, 36] evaluate the model performance relying on labeled target data, which is not always available in real-world scenarios. Therefore, we propose a Detection Adaptation Score (DAS) to evaluate the model performance in an unsupervised manner, enabling the application of DAOD models in real-world scenarios.

**Unsupervised Model Selection (UMS) for UDA.** Unsupervised Model Selection for UDA evaluates model performance in the target domain without involving annotations. Some previous works [22, 45, 40, 44, 62, 9, 26, 59] seek to predict model performance in OOD scenarios. PS [28], ATC [22] leverage the prediction confidence, and [49, 48] estimate from the perspective of entropy. DEV [61] estimates and decreases the target risk by embedding adapted feature representation while validation. TS [59] examines the spatial uniformity of the classifier, as well as the transferability and discriminability of deep representation. However, they mainly focus on classification tasks. For object detection, the BoS [60] estimates the detection performance via the stability of bounding boxes with feature dropout but does not consider the domain discrepancy, which is vital for DAOD methods. To this end, we introduce the Detection Adaptation Score (DAS), which consists of a Flatness Index Score (FIS) and a Prototypical Distance Ratio (PDR) score by seeking a flat minima model in the target domain.

# 3    Method

In the DAOD task, we are given a labeled source domain, which includes images annotated with bounding boxes and class labels, and an unlabeled target domain containing only unlabeled images. Let us denote $\mathcal{D}_s = \{(x_i^s, \mathbf{y}_i^s)\}_{i=1}^{N_s}$ drawn from a data distribution $\mathcal{P}_s$ as the labeled source domain and $\mathcal{D}_t = \{x_j^t\}_{j=1}^{N_t}$ drawn from a data distribution $\mathcal{P}_t$ as the unlabeled target domain. Distributions $\mathcal{P}_s$ and $\mathcal{P}_t$ are related but different domains, *i.e.*, $(\mathcal{P}_s \neq \mathcal{P}_t)$. In other words, they have distinct domain shifts. And $\mathbf{y}_i^s = \{(\mathbf{b}_{ij}^s, c_{ij}^s)|_{j=1}^{m_i}\}$, where $\mathbf{b}_{ij}^s \in \mathbb{R}^4$ and $c_{ij}^s \in \{1, \ldots, K\}$ are the bounding box and corresponding category for each object, and $m_i$ is the total number of objects in an image $x_i^s$. The goal of the DAOD approach is to learn an object detection model that performs well on the target domain from both the labeled source and unlabeled target domains. In this work, we consider the model selection for DAOD approaches. In particular, there are $M$ models $\mathcal{F} = \{f^l|_{l=1}^M\}$ from different epochs or iterations. Our goal is to propose a proper metric for model evaluation in an unsupervised manner that can reflect the detection performance (*i.e.*, mAP) of the DAOD models.

In the following, we first introduce the domain adaptation generalization bound with flat minima in Sec. 3.1. Then, we will introduce our Detection Adaptation Score (DAS) in detail, which consists of a Flatness Index Score (FIS) in Sec. 3.2 and a Prototypical Distance Ratio (PDR) score in Sec. 3.3.

## 3.1    Domain Adaptation Generalization Bound with Flat Minima

In this paper, we propose a novel unsupervised model selection approach for the DAOD task, which can almost select the optimal checkpoint for the target domain. Our approach is based on the assumption that flat minima exhibit better model generalization than sharp minima, which has been evidenced by many literatures [21, 7]. The model parameters at flat minima will have smaller changes of loss values within its neighborhoods than sharp minima. To find flat minima, traditional methods require labeled data, which is unrealistic for DAOD. As the target labels are unavailable for model evaluation in DAOD scenarios, we cannot directly find the flat minima. To this end, we derive a generalization error bound as follows:

**Theorem 1.** *Given any $\delta \geq 0$, exist hypothesis $h \in \mathcal{H}$ where $\mathcal{H}$ is the hypothesis set, $\theta_h$ denotes the parameters of $h$. Given any hypothesis $h' \in \{h'|h' \in \mathcal{H}, \|\theta_{h'} - \theta_h\|_2 \leq \tau\}$, which is located in the neighborhood of $h$ with radius $\tau > 0$, the following generalization error bound holds with at least a probability of $1 - \delta$,*

$$\mathcal{E}_{\mathcal{T}}(h') \leq |\mathcal{E}_{\mathcal{T}}(h') - \mathcal{E}_{\mathcal{T}}(h)| + \mathcal{E}_{\mathcal{S}}(h) + dis(\mathcal{S}, \mathcal{T}) + \Omega, \tag{1}$$

*where $dis(\mathcal{S}, \mathcal{T})$ is the distribution mismatch between the source domain $\mathcal{S}$ and target domain $\mathcal{T}$. $\mathcal{E}_{\mathcal{S}}(h)$ is the risk of $h$ on the source domain. $\Omega$ is a constant term. Proof is provided in the appendix.*

The generalization bound shows that in addition to the constant term $\Omega$, the risk for the target hypothesis $h'$ around by $h$ with $\tau$-ball radius can be bounded by three terms: the flatness, *i.e.*, the difference between the original $h$ and neighborhood $h'$ of $h$, the error on the source domain $\mathcal{D}_s$, and the distribution distance between the source and target domains. Usually, the error on source $\mathcal{E}_{\mathcal{S}}(h)$ can be minimized with the labeled source samples. To this end, one can minimize the first and third terms to find the flat minima.

From the analysis in Theorem 1, the flatness can be deemed as model variance, while the minima depends on the domain distribution distance for DAOD task. Accordingly, we propose a Flatness Index Score (FIS) to assess the flatness by measuring the classification and localization fluctuation before and after perturbations of model parameters, and a Prototypical Distance Ratio (PDR) score to seek the minima by measuring the transferability and discriminability of the models. In this way, the proposed DAS approach can effectively represent the degree of flat minima and evaluate the model generalization ability on the target domain.

## 3.2    Flatness Index Score

In this subsection, we introduce the Flatness Index Score (FIS), which assesses the flatness of the model parameters by measuring the variance in outputs before after parameter perturbations. For object detection, we calculate the variance of both classification and localization predictions. For the property of minima on the target domain, we introduce it later in Sec. 3.3.

Specifically, let $\gamma$ denote the radius of the parameter space. We can then obtain the neighbor model $f(\cdot; \theta')$ by adding a perturbation $\Delta$ to the original parameter $\theta$, $i.e.$, $\theta' \leftarrow \theta + \Delta$. We control $\gamma$ of the perturbation as a constant ($i.e.$, $\|\Delta\| = \gamma$), thus the neighbor model $f(\cdot; \theta')$ lies on a fixed radius sphere of the original model $f(\cdot; \theta)$.

We measure the prediction correspondence between the original and neighbor models. Given an input target domain image $x_i$, the original model predicts $\{(\mathbf{b}_j, \mathbf{p}_j)|_{j=1}^{n_i}\}$ as the neighbor model predicts $\{(\tilde{\mathbf{b}}_j, \tilde{\mathbf{p}}_j)|_{j=1}^{n_i'}\}$, where $\mathbf{b}_j$ is the bounding box and $\mathbf{p}_j$ is the probability vector of the $j$-th instance. We use $d_{jj'}(f_\gamma(x_i; \theta))$ to represent the matching cost of the $j$-th and $j'$-th object in two model's predictions on image $x_i$. As the model predictions has results from both regression and classification branches, the matching cost $d_{jj'}$ contains the divergence of bounding boxes and classification probability vectors as follows:

$$d_{jj'} = \mathrm{KL}(\mathbf{p}_j, \tilde{\mathbf{p}}_{j'}) - \mathrm{IoU}(\mathbf{b}_j, \tilde{\mathbf{b}}_{j'}), \tag{2}$$

where IoU is the intersection over union of two boxes, KL is the KL-divergence over two probability vectors. The smaller the $d_{jj'}$ is, the better the two object predictions match. Then the Flatness Index Score (FIS) on the original model parameter $\theta$ with radius $\gamma$ is defined as:

$$\mathrm{FIS} = -\mathbb{E}_{\theta' \leftarrow \theta + \Delta}[\mathbb{E}_{x_i \sim \mathcal{D}_t}[\min_\sigma \frac{1}{n_i''} \sum_{j=1}^{n_i''} d_{j\sigma(j)}]], \tag{3}$$

where $n_i'' = \min\{n_i, n_i'\}$ is the smaller number of the predicted objects from the original and neighbor models. $\sigma(j)$ is the $j$-th object's one-to-one corresponding index leading to the minimized matching cost when $n_i \leq n_i'$, otherwise $\mathrm{FIS} = -\mathbb{E}_{\theta' \leftarrow \theta + \Delta}[\mathbb{E}_{x_i \sim \mathcal{D}_t}[\min_\sigma \frac{1}{n_i''} \sum_{j'=1}^{n_i''} d_{\sigma(j')j'}]]$. Hungarian Algorithm is applied to select the best-matched pairs.

### 3.3 Prototypical Distance Ratio

To facilitate the search for flat minima, we explore methods to identify minima regions in the target domain. In DAOD, the model with better transferability and discriminability would perform well on the target domain. There are many methods to evaluate the domain distance between the source and target domains, for example, Maximum Mean Discrepancy (MMD) [25] and Proxy A-distance (PAD) [1]. However, simply measuring the image feature distribution is not feasible to reflect the transferability of the detection model. Meanwhile, the traditional discriminability metrics such as entropy [48], and mutual information [33] also fail to effectively correlate the model performance. In this paper, we consider the class prototype distance of instances in images across domains to evaluate the transferability and discriminability of the DAOD models. The class prototype of instances is the center of a specific class and aggregates the instance information from the samples. In unsupervised domain adaptation, prototype-based domain alignment is comprehensively studied in the literature [58, 57, 43] and attempts to narrow the distance between prototypes of the same categories of two domains, showcasing the effectiveness of prototype alignment.

Now, we show how to leverage prototype distance to effectively evaluate the transferability and discriminability of the DAOD models. Our prototype is calculated based on the instance feature, $e.g.$, the proposal feature in Faster RCNN. In particular, we denote the instance feature as $F_{ij} \in \mathbb{R}^d$ for the $j$-th instance in the $i$-th image and the final classification probability vector as $\mathbf{p}_{ij}$. The prototype of the $k$-th class $P_k \in \mathbb{R}^d$ for the target domain can be calculated softly as follows:

$$P_k = \mathbb{E}_{x_i \sim \mathcal{D}_t}[\frac{1}{n_i^\mathrm{p}} \sum_{j=1}^{n_i^\mathrm{p}} F_{ij} \cdot \mathbf{p}_{ij}^k], \tag{4}$$

where $n_i^\mathrm{p}$ is the number of proposals in the $i$-th image. The source prototypes can be obtained similarly. Finally, we have the source and target domain prototypes $P^\mathcal{S} \in \mathbb{R}^{K \times d}$ and $P^\mathcal{T} \in \mathbb{R}^{K \times d}$.

An ideal DAOD network aligns the features of the same category between domains while increasing the feature distances between different categories. We propose a Prototypical Distance Ratio (PDR) score to evaluate this ability with the category-wise prototypes. The prototype distance of the same

category across domains can be defined as follows:

$$d_{\text{intra}} = \frac{1}{K} \text{trace}(M(P^{\mathcal{S}}, P^{\mathcal{T}})), \tag{5}$$

where $M(P^{\mathcal{S}}, P^{\mathcal{T}}) \in \mathbb{R}^{K \times K}$ is the category-wise distance matrix between $P^{\mathcal{S}}$ and $P^{\mathcal{T}}$. Denote $M_{kk'}(P, P')$ as the distance between $P_k$ and $P'_{k'}$.

Similarly, we could obtain the inter-category prototype distances as follows:

$$d_{\text{inter}} = \delta(P^{\mathcal{S}}, P^{\mathcal{T}}) \cdot \delta(P^{\mathcal{S}}, P^{\mathcal{S}}) \cdot \delta(P^{\mathcal{T}}, P^{\mathcal{T}}), \tag{6}$$

where $\delta(P, P') = \frac{1}{K^2 - K} \sum_{k \neq k'}^{K} M_{kk'}(P, P')$ is a function to calculate distance among different categories. Combine the intra-category distance and inter-category distance into our Prototypical Distance Ratio (PDR) score as follows:

$$\text{PDR} = d_{\text{inter}} / d_{\text{intra}}. \tag{7}$$

The proposed PDR score can be an effective metric to evaluate the transferability and discriminability of the DAOD models. The higher PDR score indicates that the instance features of detection models have better properties with large inter-category distances and small intra-category distances.

### 3.4 Detection Adaptation Score

For different checkpoints in a training session, we simply normalize the FIS and PDR separately by min-max normalization. The DAS is a combination of FIS and PDR as follows.

$$\text{DAS}_t = \overline{\text{FIS}}_t + \lambda \overline{\text{PDR}}_t, \tag{8}$$

where $t$ is the index of the checkpoint. $\overline{\text{FIS}}_t$ and $\overline{\text{PDR}}_t$ are the $t$-th FIS and PDR score normalized among all checkpoints. $\lambda$ is a trade-off parameter that controls the contribution of $\overline{\text{PDR}}_t$ for $\text{DAS}_t$.

## 4 Experiments

### 4.1 Experiment Setup

**Benchmarks.** We follow previous works [11, 50, 38] and evaluate the effectiveness of the proposed DAS on the following adaptation scenarios:

- **Real-to-Art Adaptation (P2C)**: In this scenario, we test our proposed method with domain shift between the real image domain and the artistic image domain. Following [50], we choose the PASCAL VOC 2007/2012 and Clipart1k as the source and target domains, respectively. **PASCAL VOC** [20] is a widely used benchmark in the object detection community. We use the 2007 and 2012 versions of PASCAL VOC that contain about 15k images with instance annotations for 20 object classes. **Clipart1k** [31] contains 500 train images and 500 testing images in clipart style, annotated with bounding boxes for the same 20 object categories as in the PASCAL VOC.

- **Weather Adaptation (C2F)**: In real-world scenarios, such as autonomous driving systems, object detectors may encounter various weather conditions. We study the adaptation from normal to foggy weather. In particular, we use the Cityscapes and Foggy Cityscapes as the source and target domains, respectively. **Cityscapes** [13] contains a diverse set of urban street scenes captured from 50 cities and $2,975$ training images and $500$ validation images, annotated for 8 object classes. **Foggy Cityscapes** [51] is a variant of the Cityscapes dataset where synthetic fog is added to the images to simulate adverse weather conditions. The annotations remain consistent with the original Cityscapes dataset. Note that we choose the worst foggy level (*i.e.*, 0.02) of Foggy Cityscapes in the experiment following [11].

- **Synthetic-to-Real Adaptation (S2C)**: Synthetic images offer an alternative solution for addressing the data collection and annotation issues. However, there is a distinct distribution mismatch between synthetic images to real images. To adapt the synthetic to the real scenes, we utilize Sim10k as the source domain and Cityscapes as the target domain. **Sim10k** [32] consists of 10,000 synthetic images generated from a simulation environment, with annotations for car bounding boxes. Because only the car is annotated in both domains, we report the AP of "car" in the validation set of Cityscapes.

Table 1: The detection performance (mAP in %) comparison between the last checkpoints, the checkpoints selected by DAS, and the oracle checkpoints.

| Benchmark DAOD | Real-to-Art Adaptation | | | | Weather Adaptation | | | | Synthetic-to-Real Adaptation | | | |
|---|---|---|---|---|---|---|---|---|---|---|---|---|
| | Last | Ours | Imp.↑ | Oracle | Last | Ours | Imp.↑ | Oracle | Last | Ours | Imp.↑ | Oracle |
| DAF | 15.72 | 17.16 | +1.44 | 17.49 | 28.27 | 29.01 | +0.74 | 29.17 | 43.05 | 45.09 | +2.04 | 45.09 |
| MT | 34.25 | 35.27 | +1.02 | 35.75 | 45.74 | 45.74 | +0.00 | 47.51 | 54.85 | 55.41 | +0.56 | 55.41 |
| AT | 41.98 | 47.83 | +5.85 | 47.83 | 48.16 | 49.26 | +1.10 | 49.26 | 28.35 | 47.44 | +19.09 | 47.44 |
| CMT | 40.11 | 40.11 | +0.00 | 40.82 | 48.46 | 49.15 | +0.69 | 49.36 | 42.35 | 48.81 | +6.46 | 55.53 |

Table 2: Comparison of different unsupervised model evaluation methods on real-to-art adaptation (P2C), weather adaptation (C2F), and synthetic-to-real adaptation (S2C).

| Benchmark Methods | Real-to-Art Adaptation | | | | Weather Adaptation | | | | Synthetic-to-Real Adaptation | | | | Average |
|---|---|---|---|---|---|---|---|---|---|---|---|---|---|
| | DAF | MT | AT | CMT | DAF | MT | AT | CMT | DAF | MT | AT | CMT | |
| Last Ckpt. | 15.72 | 34.25 | 41.98 | 40.11 | 28.27 | 45.74 | 48.16 | 48.46 | 43.05 | 54.85 | 28.35 | 42.35 | 39.27 |
| PS | 15.72 | 34.25 | 43.93 | 39.93 | 28.27 | **46.95** | 48.23 | 48.46 | 42.68 | 55.29 | 32.56 | 47.18 | 40.29 (+1.02) |
| ES | **17.49** | 34.25 | 43.93 | 39.93 | 26.77 | **46.95** | 47.98 | 48.86 | 42.68 | 54.85 | 30.57 | 42.99 | 39.77 (+0.50) |
| ATC (th=.3) | 15.72 | 34.25 | 43.93 | 39.93 | 28.27 | **46.95** | 48.23 | 48.46 | 42.68 | 55.29 | 32.56 | 47.18 | 40.29 (+1.02) |
| ATC (th=.4) | 15.72 | 34.25 | 41.98 | **40.11** | 28.27 | **46.95** | 48.23 | 48.46 | 42.68 | 55.29 | 32.56 | 47.18 | 40.14 (+0.87) |
| ATC (th=.95) | 17.32 | 34.25 | 43.93 | 39.93 | **29.01** | **46.95** | 48.23 | 48.46 | 43.05 | 54.85 | 32.56 | 47.18 | 40.45 (+1.18) |
| FD | 13.50 | 31.22 | 45.18 | **40.11** | 28.27 | 45.74 | 48.69 | 40.48 | 43.05 | **55.41** | **47.44** | 42.35 | 40.12 (+0.85) |
| TS | 14.17 | 33.62 | 45.02 | 40.03 | 28.27 | **46.95** | 48.69 | **49.36** | 42.56 | 54.76 | 30.57 | 47.18 | 40.10 (+0.83) |
| BoS | 13.83 | 34.25 | 43.93 | **40.11** | 19.07 | 42.21 | 44.91 | 43.10 | 44.18 | 55.29 | 31.59 | **54.53** | 38.92 (-0.35) |
| **DAS (ours)** | 17.16 | **35.27** | **47.83** | **40.11** | **29.01** | 45.74 | **49.26** | 49.15 | **45.09** | **55.41** | **47.44** | 48.81 | **42.52 (+3.25)** |
| Oracle | 17.49 | 35.75 | 47.83 | 40.82 | 29.17 | 47.51 | 49.26 | 49.36 | 45.09 | 55.41 | 47.44 | 55.53 | 43.39 |

**DAOD Frameworks.** In this work, we test our method on four classical DAOD models in the two main DAOD streams (*i.e.*, adversarial training and self-training): DA Faster RCNN (**DAF**) [11], Mean-Teacher (**MT**) [52], Adaptive Teacher (**AT**) [38], Contrastive Mean-Teacher (**CMT**) [4]. DAF [11] is a typical DAOD framework extending the Faster RCNN architecture by incorporating domain adaptation techniques. It employs adversarial training to align the feature distribution across domains at both image and instance levels. MT [52] leverages a teacher-student paradigm to generate pseudo labels to provide extra supervision for the student model on the target domain. The teacher model is an exponential moving average (EMA) of the student model and thus can provide more accurate pseudo-labels for the student model. This approach helps in leveraging unlabeled data by enforcing consistency between the predictions of the teacher and student models. AT [38] improves upon the Mean-Teacher framework by employing an adversarial learning module to align the feature distributions across two domains, reducing domain bias and improving pseudo-label quality. CMT [4] enhances the Mean-Teacher framework with contrastive learning techniques by encouraging similar instances to be closer in the feature space while pushing the features of dissimilar instances apart.

**Unsupervised Model Selection Baselines.** Existing unsupervised model selection methods for UDA tasks are mainly based on classification tasks, such as Prediction Score (PS) [28], Entropy Score (ES) [48], Average Threshold Confidence (ATC) [22], Transfer Score (TS) [59]. We reproduce these approaches on the classification branch of object detection models. Besides, we also use the Fréchet Distance (FD) [17] to measure the domain distance on backbone features as a compared method. Bounding Box Stability (BoS) is introduced to tackle unsupervised model evaluation problems specifically for object detection networks. It evaluates the model generalization of the detection model by probing the bounding box stability under feature dropout.

**Implement Details.** Following previous works [11], we choose one of the representative detection frameworks Faster RCNN as our base detector. We followed the instructions from the released code of the DAOD methods and reproduced their results. The hyperparameters, learning rates, and optimizers are set according to the default configurations provided in the original papers. We set our hyperparameters $\lambda = 1$ and $\gamma = 1$ in all experiments, which perform well on all the benchmarks. Our implementation is built upon the Detectron2 detection framework. We have added a detailed implementation in the appendix.

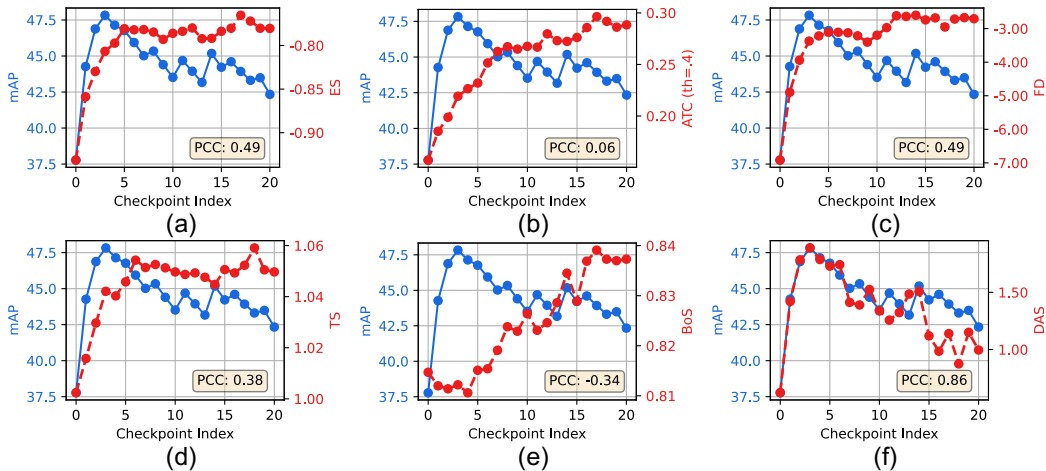

Figure 2: The comparison of different unsupervised model evaluation methods for DAOD. The experiments are conducted on real-to-art adaptation (P2C) using AT. Note that the directions of all scores are unified.

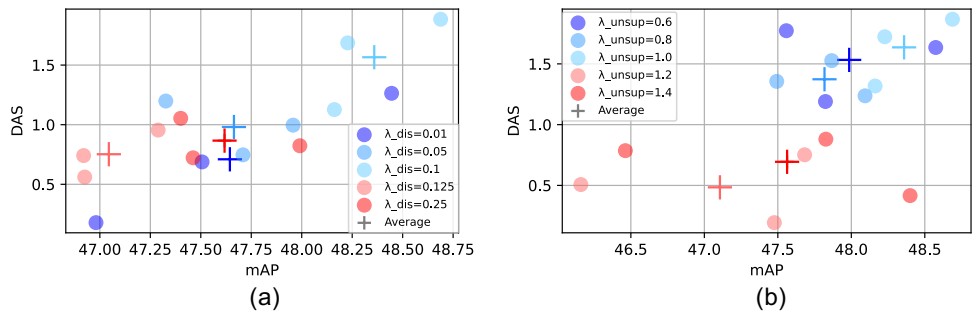

Figure 3: Hyperprameter tuning on AT [38] using our DAS. (a) $\lambda_{\text{dis}}$ that controls the weight of adversarial loss from domain discriminator. (b)$\lambda_{\text{unsup}}$ that controls the weight of unsupervised loss.

## 4.2 Main Results

**Checkpoint Selection after Training.** Checkpoint selection after training is pivotal for the application of DAOD approaches in real-world scenarios since the DAOD models usually suffer from negative transfer and overfitting the target domain during training. For example, on the real-to-art adaptation (P2C), AT [38] drops $5.85\%$ from the highest mAP ($47.83\%$) to the last checkpoint ($41.98\%$). The detailed results on checkpoint selection, with the highest DAS, after training are listed in Table 1. Compared with the last checkpoint, the proposed method works well on almost all the DAOD methods. For example, the proposed method for AT [38] achieves $5.85\%$, $1.1\%$, and $19.09\%$ improvements in terms of mAP on real-to-art, weather, and synthetic-to-real adaptations, respectively. These experimental results demonstrate that the proposed method can choose a reliable checkpoint and thus avoid the negative transfer and overfitting on the target domain during training.

Moreover, we compare our method with the recent unsupervised model evaluation methods in Table 2, showing that our method consistently outperforms all the compared baselines. For instance, our methods outperform recent works TS [59] and BoS [60] by $2.42\%$ and $3.60\%$ mAP on average, respectively. These results demonstrate the effectiveness of the proposed method in choosing the optimal checkpoints. We also present the correlation between unsupervised model evaluation metrics and the ground truth mAP (*i.e.*, using the annotations in the target domain to evaluate the model) in Fig. 2. It is clearly shown that the previous methods give higher scores as the training goes on and fail to correlate with the performance of DAOD checkpoints. In contrast, the proposed DAS score correlates well with the performance of DAOD checkpoints (*i.e.*, 0.86 PCC).

Table 3: The comparison of different methods for hyperparameter tuning on Weather Adaptation.

| Hyperparam. | | PS | ES | ATC th=.4 | FD | TS | BoS | DAS (ours) |
|---|---|---|---|---|---|---|---|---|
| $\lambda_{\text{dis}}$ | mAP | 47.99 | 47.99 | 47.99 | **48.69** | 47.99 | 47.99 | **48.69** |
| | PCC | -0.346 | -0.399 | 0.348 | 0.631 | 0.007 | 0.024 | **0.865** |
| $\lambda_{\text{unsup}}$ | mAP | 47.56 | 47.87 | 47.56 | **48.69** | 48.57 | 48.09 | **48.69** |
| | PCC | 0.273 | 0.453 | 0.298 | 0.751 | 0.557 | 0.659 | **0.938** |

Table 4: Ablation study of the proposed DAS
method. The results are averaged from DAOD
benchmarks and approaches.

| | PDR | FIS | mAP | PCC |
|---|---|---|---|---|
| Last Checkpoint | | | 39.27 | - |
| | | ✓ | 41.74 | 0.48 |
| Ours | ✓ | | 42.14 | 0.64 |
| | ✓ | ✓ | 42.52 | 0.67 |

Table 5: The hyperparameter sensitivity of the
proposed method on real-to-art adaptation.

| | $\lambda$ | mAP | PCC |
|---|---|---|---|
| Last Checkpoint | - | 33.02 | - |
| | 0.1 | 34.76 | 0.748 |
| | 0.5 | 35.08 | 0.842 |
| Ours | 1.0 | **35.09** | **0.854** |
| | 2.0 | 35.05 | 0.821 |
| | 10.0 | 35.05 | 0.729 |

**Hyperparameter Tuning for DAOD Methods.** Domain adaptation methods can be highly sensitive to the hyperparameters. Inappropriate hyperparameter selection would limit the transfer performance and lead to negative transfer, which even leads models perform below the source-only ones. Therefore, the validation of hyperparameters is an important problem in DAOD, yet researchers have unfortunately overlooked it. We evaluate our DAS in hyperparameters tuning by leveraging the weights of adversarial loss (denoted as $\lambda_{\text{dis}}$) for the domain discriminator and of unsupervised loss (denoted as $\lambda_{\text{unsup}}$) for self-training in AT [38]. For a model with a specific hyperparameter setting, we obtain the last three checkpoints of its training process and calculate their average performance to represent the model. The final DAS of the specific model is defined as the average DAS of these obtained checkpoints. The experimental results are conducted on the weather adaptation and summarized in Fig. 3. From Fig. 3, we can observe that the model with higher DAS presents better improvement after adaptation. The proposed DAS can correlate the detection performance without any labels on the target domain. We further compare our DAS with the unsupervised model evaluation methods and summarize the results in Table 3. The proposed DAS consistently outperforms the previous baselines, which demonstrates the effectiveness of the proposed method in hyperparameter tuning. In a nutshell, our DAS can be a good indicator to guide the hyperparameter tuning for DAOD methods, thus avoiding the negative model optimization due to the inappropriate hyperparameter selection.

### 4.3 Further Analysis

**Ablation Study.** We conduct the ablation study of the proposed method by isolating DAS into separate metrics. In particular, we conduct experiments on three benchmarks and use four DAOD methods including DAF [11], MT [52], AT [38], and CMT [4]. We summarize the results in Table 4. The experimental results indicate that the proposed PDR score and FIS evaluate the model performance without labels effectively. In particular, the PDR score and FIS select checkpoints with mAP $42.14\%$ and $41.74\%$, PCC $0.64$ and $0.48$, respectively. DAS combines them and further improves their performance to $42.52\%$ mAP and $0.67$ PCC, demonstrating the synergy effect among them.

**Hyperparameter Sensitivity.** We investigate the sensitivity of the hyperparameter $\lambda$ controlling the weight of PDR for DAS on real-to-art adaptation (P2C). The results are summarized in Table 5. $\lambda = 1.0$ achieves the best average results. From a wide range of $\lambda$, DAS has relatively stable results.

## 5 Conclusion

In this work, we propose a novel metric named Detection Adaptation Score (DAS) by seeking flat minima to evaluate the domain adaptive object detection models without involving any target labels. The proposed DAS consists of a Flatness Index Score (FIS) and a Prototypical Distance Ratio (PDR) score and can find flat minima of DAOD models in an unsupervised way. Extensive experiments have been conducted on public DAOD benchmarks for several classical DAOD methods. Experimental results indicate that the proposed DAS correlates well with the performance of the detection model and thus can be used for checkpoint selection after training. The proposed method fosters the application of DAOD methods in the real-world scenario. We hope that our work will inspire researchers and contribute to advancing research in DAOD.

# Acknowledgments and Disclosure of Funding

This work is supported by the National Natural Science Foundation of China (No. 62176047), the Sichuan Science and Technology Program (No. 2022YFS0600), Sichuan Natural Science Foundation (No. 2024NSFTD0041), and the Fundamental Research Funds for the Central Universities Grant (No. ZYGX2021YGLH208).

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

# A Appendix

In this appendix, we first give the proof of Theorem 1 in Sec. A.1. And then provide more implementation details in Sec. A.2 and experimental results in Sec. A.3. Finally, we discuss the limitations and societal impact of our work.

## A.1 Proofs of Theorem 1

**Theorem 1.** Given any $\delta \geq 0$, exist hypothesis $h \in \mathcal{H}$ where $\mathcal{H}$ is the hypothesis set, $\theta_h$ denotes the parameters of $h$. Given any hypothesis $h' \in \{h'|h' \in \mathcal{H}, \|\theta_{h'} - \theta_h\|_2 \leq \tau\}$ located in the neighborhood of $h$ with radius $\tau > 0$, the following generalization error bound holds with at least a probability of $1 - \delta$,

$$\mathcal{E}_{\mathcal{T}}(h') \leq |\mathcal{E}_{\mathcal{T}}(h') - \mathcal{E}_{\mathcal{T}}(h)| + \mathcal{E}_{\mathcal{S}}(h) + dis(\mathcal{S}, \mathcal{T}) + \Omega, \tag{9}$$

where $dis(\mathcal{S}, \mathcal{T})$ is the distribution mismatch between the source domain $\mathcal{S}$ and target domain $\mathcal{T}$. $\mathcal{E}_{\mathcal{S}}(h)$ is the risk of $h$ on the source domain. $\Omega$ is a constant term.

*Proof.* Given any $\delta \geq 0$, exist hypothesis $h \in \mathcal{H}$ where $\mathcal{H}$ is a hypothesis set, $\theta_h$ denotes the model parameters of $\overline{h}$. Give any hypothesis $h' \in \{h'|\|\theta_{h'} - \theta_h\|_2 \leq \tau\}$, $\tau > 0$ is the radius of the neighborhood. The following generalization bound holds with at least a probability of $1 - \delta$,

$$\begin{aligned} \mathcal{E}_{\mathcal{T}}(h') &= \mathcal{E}_{\mathcal{T}}(h') - \mathcal{E}_{\mathcal{T}}(h) + \mathcal{E}_{\mathcal{T}}(h) \\ &\leq |\mathcal{E}_{\mathcal{T}}(h') - \mathcal{E}_{\mathcal{T}}(h)| + \mathcal{E}_{\mathcal{T}}(h) \end{aligned} \tag{10}$$

According to the domain adaptation theory from [34, 64], we have the following inequality:

$$\mathcal{E}_{\mathcal{T}}(h) \leq \mathcal{E}_{\mathcal{S}}(h) + dis(\mathcal{S}, \mathcal{T}) + \Omega, \tag{11}$$

where the $\mathcal{E}_{\mathcal{S}}(h)$ indicates the source error and $dis(\mathcal{S}, \mathcal{T})$ denotes the distribution mismatch between the source domain $\mathcal{S}$ and the target domain $\mathcal{T}$. In this way, we replace the last term $\mathcal{E}_{\mathcal{T}}(h)$ in Eq. (10) with Eq. (11), leading to the following error bound:

$$\mathcal{E}_{\mathcal{T}}(h') \leq |\mathcal{E}_{\mathcal{T}}(h') - \mathcal{E}_{\mathcal{T}}(h)| + \mathcal{E}_{\mathcal{S}}(h) + dis(\mathcal{S}, \mathcal{T}) + \Omega, \tag{12}$$

Till now, we prove the inequality in Theorem 1. $\qquad\square$

## A.2 More Implementation Details

**Training and Evaluating Details.** In our experiment, all the DAOD models are trained according to the default setting specified in their original papers and the open-source codebases. For DA Faster RCNN, we train it on the three benchmarks with a VGG-16 [12] backbone pre-trained on ImageNet, using a batch size of 4. For Mean Teacher, Adaptive Teacher, and Contrastive Mean Teacher, we train the models with a pre-trained ResNet-101 [27] backbone on ImageNet for the real-to-art adaptation setting, and with a pre-trained VGG-16 [12] backbone for the weather adaptation and synthetic-to-real adaptation settings. The batch size of AT, MT, and CMT are set to 8. Training is conducted on four RTX 3090 GPUs or two A100 GPUs according to the computational requirements of different DAOD methods. For all the results, we report the mean Average Precision (mAP) at the IoU threshold of 0.5.

**Detection Adaptation Score.** For the Flatness Index Score (FIS), we perturb the detector model by adding a normalized random direction for all the model parameters including the backbone and detection head. For the Prototypical Distance Ratio (PDR) score, we use the features of the region proposals and the corresponding predicted probability (including background) from the classification branch of the detection head to aggregate the information. In our implementation, $L2$ distance is used as the prototype distance.

## A.3 More Experimental Results

**Top3 Checkpoint Selection Results after Training** We summarize the top3 checkpoint selection results in Table 6. We can observe that with more loose selection condition, our method still outperforms other unsupervised model selection methods by a large margin. These results further demonstrate the effectiveness of the proposed DAS in select optimal checkpoints for DAOD methods.

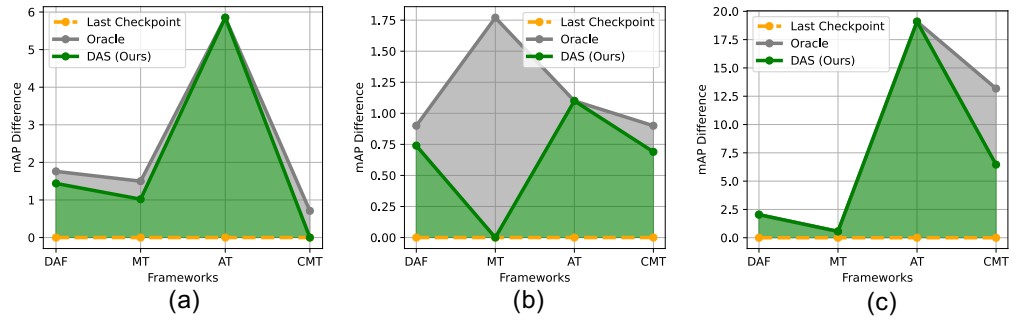

Figure 4: The performance gap among the last checkpoint, our DAS, and Oracle checkpoint. (a) real-to-art adaptation, (b) the weather adaptation, and (c) the synthetic-to-real adaptation.

**Performance Gap between Last Checkpoint and Oracle Model.** In Fig. 4, we compare the DAS selection results between the last checkpoint and the Oracle checkpoint (*i.e.*, using the ground-truth labels in the target domain to evaluate the model performance) of various frameworks and benchmarks. It shows that there is a significant performance gap between them, making it urgent to design an effective method for unsupervised model selection in DAOD scenarios. In this paper, we propose DAS by seeking the flat minima of the DAOD models. From Fig. 4, we can observe that the proposed DAS shows promising unsupervised model selection results. While our method has made progress compared to others, there is still room for further improvement.

**More Visual Correlation Results.** We show the correlation comparison results in Fig. 5 under the real-to-art adaptation using MT [52]. Overall, our DAS correlates well with the detection performance and outperforms the previous method. It is worth noting that the compared method cannot figure out the overfitting issue while our method successfully reflects it.

**More Unsupervised Model Selection Results**. The DAS can select models on other DA methods and tasks. 1) We also conducted our method on SIGMA++ [37] on weather adaptation, which minimizes the domain gap by graph matching. DAS chooses a checkpoint at $41.7\%$ mAP (which is the oracle), while the last checkpoint reaches $39.5\%$ mAP. 2) For prototype-based DAOD methods. The DAS includes FIS and PDR derived from the generalization bound. It can evaluate DAOD models from different aspects. Some existing works like GPA [58] use the prototype-based alignment method to minimize the domain gap between domains. However, the prototype estimation in existing works only utilizes samples in a mini-batch during the model training. In contrast, our DAS uses the entire dataset to estimate the prototypes, which is more robust and has better generalization ability. To verify this, we experimented on Synthetic-to-Real adaptation for GPA [58]. The DAS chooses a checkpoint at $45.8\%$ mAP (the oracle is $47.0\%$) while the last checkpoint reaches $43.1\%$. Our proposed method still works when DAOD frameworks also optimize the prototype-based distance. 3) For other DA tasks. Our method can also work on other DA tasks, such as semantic segmentation. We conducted our DAS on the well-known DAFormer [29] method on GTA5 to Cityscapes Adaptation. It is shown that our DAS can choose a better checkpoint of the model with an average mIoU of $64.2\%$ while the last checkpoint of the model only achieves $60.9\%$ and the oracle checkpoint is at $65.9\%$.

## A.4 Limitation

Although our method outperforms many unsupervised model evaluation methods in many DAOD benchmarks and methods, it still has considerable room for improvement. There are certain scenarios where our proposed method faces limitations, such as the inability to consistently select the best checkpoint and a lack of sufficient correlation between the proposed DAS and the ground truth detection performance of DAOD models on the target domain. To address these limitations, we could consider incorporating a more fine-grained distribution alignment metric to evaluate the distance across domains. On the other hand, we can explore methods that extend beyond the constraints of labeled data by leveraging few-shot labeled data in the target domain to help model evaluation. Another limitation is that our method requires the entire dataset of the source and target domains to calculate the evaluation score. It would be beneficial to develop a more data-efficient approach that can effectively evaluate the performance of DAOD models while minimizing the data requirements.

Table 6: The top3 selection results of different unsupervised model evaluation methods on real-to-art adaptation (P2C), weather adaptation (C2F), and synthetic to real adaptation (S2C).

| Benchmark | Real-to-Art Adaptation | | | | Weather Adaptation | | | | Synthetic-to-Real Adaptation | | | | Average |
|---|---|---|---|---|---|---|---|---|---|---|---|---|---|
| Methods | DAF | MT | AT | CMT | DAF | MT | AT | CMT | DAF | MT | AT | CMT | |
| Last Ckpt. | 15.72 | 34.25 | 41.98 | 40.11 | 28.27 | 45.74 | 48.16 | 48.46 | 43.05 | 54.85 | 28.35 | 42.35 | 39.27 |
| PS | 17.32 | 34.25 | 43.93 | 40.11 | 29.17 | 47.51 | 48.69 | 48.46 | 43.05 | 55.29 | 32.56 | 50.43 | 40.90 |
| ES | 17.49 | 34.25 | 43.95 | 40.11 | 29.01 | 46.95 | 48.69 | 49.15 | 44.33 | 55.29 | 32.56 | 44.66 | 40.54 |
| ATC (th=.3) | 17.32 | 34.25 | 43.93 | 40.11 | 29.17 | 47.51 | 48.69 | 48.46 | 43.05 | 55.29 | 32.56 | 50.43 | 40.90 |
| ATC (th=.4) | 17.49 | 34.25 | 43.93 | 40.11 | 29.17 | 47.51 | 48.69 | 48.46 | 43.05 | 55.29 | 32.56 | 50.43 | 40.91 |
| ATC (th=.95) | 17.32 | 34.25 | 43.93 | 40.11 | 29.01 | 46.95 | 48.69 | 48.49 | 43.05 | 55.29 | 32.56 | 50.43 | 40.84 |
| FD | 14.70 | 33.93 | 45.18 | 40.11 | 29.17 | 47.51 | 48.69 | 48.46 | 45.09 | 55.41 | 47.44 | 44.66 | 41.70 |
| TS | 17.16 | 34.92 | 45.94 | 40.05 | 29.17 | 47.51 | 48.69 | 49.36 | 44.73 | 55.14 | 31.17 | 50.43 | 41.19 |
| BoS | 16.38 | 34.25 | 43.93 | 40.11 | 22.01 | 44.74 | 49.26 | 48.31 | 45.07 | 55.29 | 32.56 | 55.21 | 40.59 |
| DAS (ours) | 17.49 | 35.27 | 47.83 | 40.11 | 29.01 | 47.51 | 49.26 | 49.15 | 45.09 | 55.41 | 47.44 | 50.43 | **42.83** |

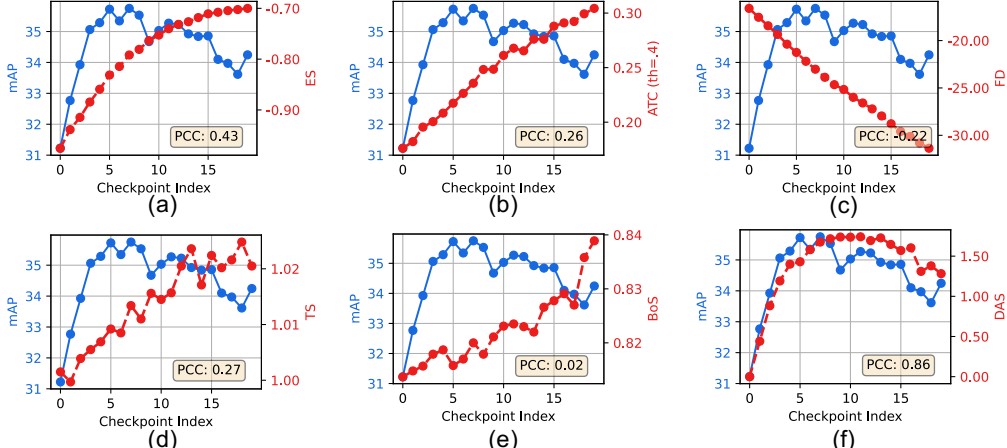

Figure 5: More comparisons on different unsupervised model evaluation methods for DAOD. The experiments are conducted on real-to-art adaptation (P2C) using MT. The direction of all the scores is unified.

## A.5  Societal Impact

Our Detection Adaptation Score (DAS) method has the potential to positively impact various downstream systems, such as autonomous driving and embodied AI. These systems frequently encounter previously unseen domains in real-world scenarios, and our method effectively addresses the unsupervised evaluation problem of domain adaptive object detection methods. However, it is important to acknowledge that the application of our method may also introduce potential negative impacts. For example, when applied to surveillance videos and medical images, privacy concerns may arise. It is essential to understand that these issues are not inherent to the technology itself but rather depend on the responsible and ethical use by human beings.

