# OpenReview forum: "Towards Unsupervised Model Selection for Domain Adaptive Object Detection"
_NeurIPS.cc/2024/Conference — NeurIPS 2024 poster_

### Official Review · Reviewer_XtC2 · 2024-07-01

**Soundness:** 3
**Presentation:** 3
**Contribution:** 3
**Rating:** 7
**Confidence:** 5

**Summary:**

The paper introduces the Detection Adaptation Score (DAS), an unsupervised approach for selecting optimal models in Domain Adaptive Object Detection (DAOD) without target domain labels. DAS is based on the flat minima principle and includes the Flatness Index Score (FIS) and Prototypical Distance Ratio (PDR) to evaluate model generalization. Experiments on various benchmarks demonstrate DAS's effectiveness in checkpoint selection and hyperparameter tuning, outperforming existing methods. The perspective of enhancing model selection for DAOD is interesting and novel. It attempts to tackle an overlooked yet important limitation of existing DAOD methods and shows potential for other computer vision tasks.

**Strengths:**

1. The paper writing is clear and easy to follow. The motivation is well described.
2. The perspective about model selection for DAOD is interesting because it seems to be the first attempt in this field. Besides, this can be extended to other communities since most tasks need to decide which checkpoint is optimal for practical usage without labels.
3. The proposed method can be deployed for different DAOD methods and give moderate gains without sacrificing inference speed.

**Weaknesses:**

1. It seems that the proposed method is only tested on the Faster RCNN baseline.
2. The performance gains on some detectors under some settings are limited.
3. The method is only tested on object detection, lacking an analysis for more fine-grained tasks, e.g., instance segmentation.

**Questions:**

I didn't find major issues with the paper, and I only have some minor confusion, as listed below.

1. Can the proposed method be extended to other baselines for DAOD, such as the FCOS-based method [1] and the DETR-based method [2]? It will be great to enhance the adaptability of the method further.

2. Could the authors explain why DAS does not give obvious gains in some settings? For example, DAS gives no gains for CMT on Real-to-Art Adaptation in Table 1.

3. Can the method be used for fine-grained tasks, such as segmentation?

4. Consider giving a more comprehensive review of some advanced cross-domain object detection works [1-4].

[1] Deng, J., Xu, D., Li, W., & Duan, L. (2023). Harmonious teacher for cross-domain object detection. In Proceedings of the IEEE/CVF Conference on Computer Vision and Pattern Recognition (pp. 23829-23838).

[2] Huang, W. J., Lu, Y. L., Lin, S. Y., Xie, Y., & Lin, Y. Y. (2022, July). AQT: Adversarial Query Transformers for Domain Adaptive Object Detection. In IJCAI (pp. 972-979).

[3] Zhou, W., Fan, H., Luo, T., & Zhang, L. (2023). Unsupervised domain adaptive detection with network stability analysis. In Proceedings of the IEEE/CVF International Conference on Computer Vision (pp. 6986-6995).

[4] Li, W., Liu, X., & Yuan, Y. (2023). Sigma++: Improved semantic-complete graph matching for domain adaptive object detection. IEEE Transactions on Pattern Analysis and Machine Intelligence, 45(7), 9022-9040.

**Limitations:**

No potential negative societal impact.

---

> ### Author Rebuttal · Authors · 2024-08-07
>
> Thank the Reviewer XtC2 for your constructive comments and valuable feedback. We appreciate the recognition of our work on motivation and perspective. Now we answer the question raised by the reviewer as follows.
>
> **Q1**: It seems that the proposed method is only tested on the Faster RCNN baseline.
>
> **A1**: We thank the reviewer for the valuable comments. To further verify the effectiveness of the proposed DAS method on other detectors, we conduct experiments on SIGMA++[4] (FCOS-based) and  AQT[2] (DETR-based). The results on Cityscapes to Foggy Cityscapes are shown below table. It is shown that the proposed method also works with the FCOS-based, and DETR-based methods. In particular, our method chooses a SIGMA++ checkpoint with 41.7% map while the last checkpoint is 39.5% mAP, demonstrating the adaptability of the proposed method on other detectors. Although DAS gives a moderate gain compared with the last checkpoint of AQT, our method selected the optimal checkpoint. We will evaluate our method on other detectors in future research.
>
> | Method  | Last | Ours | Imp.$\uparrow$ | Oracle |
> | ------- | ---- | ---- | -------------- | ------ |
> | SIGMA++ | 39.5 | 41.7 | 2.2            | 41.7   |
> | AQT     | 45.5 | 45.7 | 0.2            | 45.7   |
>
> **Q2**: Could the authors explain why DAS does not give obvious gains in some settings? For example, DAS gives no gains for CMT on Real-to-Art Adaptation in Table 1.
>
> **A2**: The main reason lies in that the performance gap between the last checkpoint and Oracle results is small, so give our method relatively small improvement space. For example, the average performance gap of AT over three settings is $8.68%$, while that of MT over three settings is $1.28%$. Nevertheless, the proposed method can select better checkpoints in most cases and at least the last checkpoints in very few cases.
>
> **Q3**: Can the method be used for fine-grained tasks, such as segmentation?
>
> **A3**: Yes, our method can be used for fine-grained tasks. To verify this, we conduct the experiment on the semantic segmentation. In particular, we evaluate the well-known domain adaptive semantic segmentation approach DAFormer[5]. The qualitative results on GTA5->Cityscapes are shown below. It showes that our DAS method can choose a better checkpoint of the model with an mIoU of 64.2% while the last checkpoint of the model only achieves 60.9%. This demonstrates that our DAS method still works effectively on domain adaptive semantic segmentation tasks.
>
> | Method   | Last | Ours | Imp.$\uparrow$ | Oracle |
> | -------- | ---- | ---- | -------------- | ------ |
> | DAFormer | 60.9 | 64.2 | 3.3            | 65.9   |
>
> **Q4**: Consider giving a more comprehensive review of some advanced cross-domain object detection works [1-4].
>
> **A4**: Thanks for the valuable feedback. We will enhance the final paper by providing a more comprehensive review of some advanced cross-domain object detection works.
>
> [1] Deng, J., Xu, D., Li, W., & Duan, L. (2023). Harmonious teacher for cross-domain object detection. In Proceedings of the IEEE/CVF Conference on Computer Vision and Pattern Recognition (pp. 23829-23838).
>
> [2] Huang, W. J., Lu, Y. L., Lin, S. Y., Xie, Y., & Lin, Y. Y. (2022, July). AQT: Adversarial Query Transformers for Domain Adaptive Object Detection. In IJCAI (pp. 972-979).
>
> [3] Zhou, W., Fan, H., Luo, T., & Zhang, L. (2023). Unsupervised domain adaptive detection with network stability analysis. In Proceedings of the IEEE/CVF International Conference on Computer Vision (pp. 6986-6995).
>
> [4] Li, W., Liu, X., & Yuan, Y. (2023). Sigma++: Improved semantic-complete graph matching for domain adaptive object detection. IEEE Transactions on Pattern Analysis and Machine Intelligence, 45(7), 9022-9040.
>
> [5] Hoyer L, Dai D, Van Gool L. Daformer: Improving network architectures and training strategies for domain-adaptive semantic segmentation[C]//Proceedings of the IEEE/CVF conference on computer vision and pattern recognition. 2022: 9924-9935.

---

> > ### Comment · Reviewer_XtC2 · 2024-08-10
> >
> > Thank you for the clarification and effort. I have read all the author's responses and other reviews, which addressed my concerns and highlighted additional strengths and potential in tasks beyond DAOD, such as segmentation and DG. Therefore, I am happy to raise my score.

---

> > > ### Author Response · Authors · 2024-08-10
> > >
> > > Thanks for your timely responses. We sincerely appreciate your constructive comments and valuable feedback, which enhance the quality of our paper. We will revise our paper in the final paper as you suggested.

---

### Official Review · Reviewer_9Lzv · 2024-07-08

**Soundness:** 2
**Presentation:** 3
**Contribution:** 2
**Rating:** 3
**Confidence:** 4

**Summary:**

This submission provides a strategy to select an appropriate domain adaptive object detection model without access to labels of target domain. The basic premise is the minima flat to instruct the selection, because minima flat means better generalization. Experiment indicate the effect of the design based on their detection adaption score.

**Strengths:**

1. This paper is easy to read and the idea is clearly presented.

2. The proposed strategy is based flat minima which is therefore rational.

3. Experiments show the effectiveness of the proposed strategy.

**Weaknesses:**

1. This paper may neglect a fact that in real application, target labels are often known for testing the provided model, and in fact the training labels may not be accessible. So, I do not think this strategy is really necessary.

2. This paper is essentially a DG domain generalizationt task, where target domain is unseen. So we can only select (train) a better model by using existing source domain data. Flat minima has also been used in DG method to instruct how to train a domain generalization model.

3. The proposed selection strategy is orthogonal to the high-level task, such as object detection. Therefore, why focus only on the domain adaptive object detection task? More tasks should also be tested such as image classification, semantic segmentation and others.

4. In experimental comparison, the authors compare the last checkpoint, which may not be fair.

5. Lacking comparisons to many SOTA DAOD models.

6. What is the computational overhead?

**Questions:**

1. It is similar to DG and therefore the proposed strategy is not novel athought the motivation sounds good.
2. Necessity of such motivation.
3. More experiments on other related tasks.

**Limitations:**

1. The motivation is interesting but does not well match real-world applications. Test labels should be seen even not too much. Otherwise, it is just a domain generalization scenario, which means the novelty of flat minima is limited.

---

> ### Author Rebuttal · Authors · 2024-08-07
>
> Thank Reviewer 9Lzv for the constructive comments. We appreciate the positive comments including clear presentation, rational strategy, and effective experiments. Following are the responses to the reviewer's concerns.
>
> **Q1**: This paper may neglect a fact that in real application, target labels are often known for testing the provided model, and in fact the training labels may not be accessible. So, I do not think this strategy is really necessary.
>
> **A1**: The goal of unsupervised domain adaptation (UDA) is to solve the scenario where the target domain labels are inaccessible, which are aligned with the real-world application. However, how to evaluate the UDA model is a long-standing problem in the UDA research community. The researcher in the UDA community uses the target domain label to select models, which is inevitable to overfit the test set. Meanwhile, it violates the assumption of unavailability of the target domain labels. The unsupervised model selection of UDA is also discussed in many previous works [16,51]. In our paper, we are the first attempt to address the unsupervised model selection for domain adaptive object detection. Moreover, because the annotation of object detection is hard to obtain, our method is meaningful for the real-world application. We believe our work would inspire researchers in the community and push the real-world application of DAOD models forward.
>
> **Q2**: This paper is essentially a DG domain generalization task, where target domain is unseen. So we can only select (train) a better model by using existing source domain data. Flat minima has also been used in DG method to instruct how to train a domain generalization model.
>
> **A2**: Our paper is intrinsically different from the DG task. Domain generalization addresses the issue where the target domain is not accessible, while our paper addresses the model selection method for DAOD. The flat minima is a general concept in machine learning and has been used in many fields. Different fields have different estimation methods. In domain generalization, flat minima are estimated using the source domain labels. However, the target domain labels are not available for domain adaptation. Therefore, we propose a novel method via a generalization bound to obtain the flat minima without using target labels to address the unsupervised model selection problem for DAOD. This is essentially different from the DG task.
>
> **Q3**: Experiments on other related tasks.
>
> **A3**: We thank the reviewer for the valuable feedback. To further verify the effectiveness of the proposed method, we conduct the experiments on the semantic segmentation. In particular, we evaluate the well-known domain adaptive semantic segmentation approach DAFormer. The qualitative results are shown below. It is shown that our DAS method can choose a better checkpoint of the model with an mIoU of 64.2% while the last checkpoint of the model only achieves 60.9%. This demonstrates that our DAS method still works effectively on domain adaptive semantic segmentation task. We will evaluate our method on more tasks in future research.
>
> | Method   | Last | Ours | Imp.$\uparrow$ | Oracle |
> | -------- | ---- | ---- | -------------- | ------ |
> | DAFormer | 60.9 | 64.2 | 3.3            | 65.9   |
>
> **Q4**: In experimental comparison, the authors compare the last checkpoint, which may not be fair.
>
> **A4**: Due to the unavailability of target labels, choosing the last checkpoint without a specifically designed model selection is a good choice. To this end, we choose the last checkpoint as a baseline. We also compare with other model selection methods in Table 2 in the manuscript to demonstrate the effectiveness of the proposed method.
>
> **Q5**: Lacking comparisons to many SOTA DAOD models.
>
> **A5**: Thank the reviewer for the comments. Our method is orthogonal to existing DAOD models. To evaluate the effectiveness of the proposed method, we evaluate our method on classic DAOD methods. To further verify the generalization of our method on SOTA DAOD models, we evaluate the model on SIGMA++. The results are shown in the below table. We can see that the proposed method also works with SIGMA++ where we choose a checkpoint with $41.7%$ AP, while the last checkpoint is $39.5%$. We will include the new results in our final paper. We will evaluate our method on more SOTA DAOD models in future research.
>
> | Method  | Last | Ours | Imp.$\uparrow$ | Oracle |
> | ------- | ---- | ---- | -------------- | ------ |
> | SIGMA++ | 39.5 | 41.7 | 2.2            | 41.7   |
>
> **Q6**: What is the computational overhead?
>
> **A6**: In the implementation, the calculation of DAS needs only two inferences for each target domain sample and one inference for each source domain sample. Thus one checkpoint requires just a few minutes to obtain its DAS. For example, we evaluate one model like AT using 2 minutes (on RTX 3090) while the model training time is usually around 30 hours. Therefore, the computational overhead is small compared with the model training. Once a model is selected by our method, the computational overhead is the same as the original detector.

---

> > ### Comment · Reviewer_9Lzv · 2024-08-10
> > **impractical and insufficient evaluation on the confidence**
> >
> > Thanks for the authors' feedback. Although the authors would like to present a strategy for model selection, I do think this is impractical in real application and the evaluation is also insufficient to support the conclusion.
> >
> > (1) In real application, for selecting or testing a model, some labeled target data from one or more target domains should be tested to guarantee the reliability of the deployed model. From this view, the proposed work may be useless.
> >
> > (2) From another view, if you think in real application the selected model "must be" reliable without need of any labeled target domain data, I think the current evaluation experiments are very insufficient. At least, you should test your selected model on more target domain data to make sure if your selected model is really reliable in real application even without using any labeled target data.

---

> > > ### Author Response · Authors · 2024-08-12
> > >
> > > Thank you for your comments, and we would like to further clarify a few points as follows.
> > >
> > > First, it is true that testing with labeled data is the ideal way to evaluate a model. However, annotating target domain data is time-consuming and labor-intensive, and the target domain labels are often unavailable, especially when the target domain changes rapidly as pointed out by Reviewer 4YNt. Therefore, how to effectively evaluate the domain adaptation models in an unsupervised way is important for real-world applications. In our work, we address the UMS problem for DAOD through the flat minima principle. Different from the traditional flat minima methods that require labeled data to evaluate how well a model is located at the flat minima region, we show via the generation bound that the flatness can be deemed as model variance while the minima depends on the domain distribution distance for the DAOD task. We believe our work can inspire other researchers in the community and make the DAOD methods more practical in real-world applications.
> > >
> > > Second, regarding the experiments, we evaluate the proposed method on the widely-used DAOD benchmark datasets of multiple domain adaptation scenarios, including real-to-art adaptation, weather adaptation, and synthetic-to-real adaptation over serval representative DAOD methods. We also compare many model selection baselines and the proposed method outperforms these methods, which demonstrates the effectiveness of our method on unsupervised model selection. We appreciate your comments for further experiments on more target domain data and sincerely look forward to your specific suggestions for the experimental design. We would like to include these results in the final version to strengthen our work.
> > >
> > > We thank the reviewer for the opportunity to further clarify the motivation behind our work. We sincerely hope that our response can address the concerns from the reviewer and the reviewer can re-evaluate this paper based on our response.

---

> > > > ### Comment · Reviewer_9Lzv · 2024-08-13
> > > > **The setting may be useless and why not do general machine learning task**
> > > >
> > > > From the feedback of the authors and the reviewer 4YNt, I may maintain the score because this strategy can not replace the target data for real application. If you would like to make sure your selected model is blind good to the real test scene (this test scene may not be unique), you should test your method in more target domains once your model is fixed. I believe that the performance for different target domains may not be good, which also demonstrate this work may be useless to some extent. In UDA or DG scenarios, the target domain or test data should be used, but for real application, more test data of different distributions should be used.
> > > > Additionally, this submission should also be used for general machine learning problems by supposing test data to be unavailable, rather than UDA, a sub branch of ML.

---

> > > > > ### Author Response · Authors · 2024-08-14
> > > > >
> > > > > Thanks for the timely response.
> > > > >
> > > > > We would like to clarify that this work does not focus on replacing the target data in real-world applications. In contrast, our work focuses on how to achieve unsupervised model selection for domain adaptive object detection (DAOD). The goal of unsupervised domain adaptation (UDA) is an important and widely studied research problem in the machine learning community (which was also pointed out by Reviewer XtC2). We believe the unsupervised model selection approach proposed in this work would be interesting to researchers in the UDA field, especially to those working on DAOD.
> > > > >
> > > > > We appreciate the suggestions on testing on the domain generalization setting. We would like to clarify that
> > > > > DG is often treated as a different task from UDA, for which the models and benchmark datasets are different. While testing on the DG setting
> > > > > is beyond the scope of this paper, we agree it is an interesting idea, and would like to explore this direction in the future. We sincerely look forward to your recognition on the motivation and contributions of this work.

---

> ### Comment · Reviewer_4YNt · 2024-08-11
>
> > (1) In real application, for selecting or testing a model, some labeled target data from one or more target domains should be tested to guarantee the reliability of the deployed model. From this view, the proposed work may be useless.
>
> I agree with you that target domain labels are necessary to give any kind of guarantees on the target domain. However, this might simply not be feasible, for example if the target domain is continuously shifting faster than labels can reasonably be obtained. In this setting, this method would select a best-effort model without any kind of guarantees, which is still useful for many real applications. I don't feel like the authors overclaim by suggesting guaranteed reliability on the target dataset.
>
> > (2) From another view, if you think in real application the selected model "must be" reliable without need of any labeled target domain data, I think the current evaluation experiments are very insufficient. At least, you should test your selected model on more target domain data to make sure if your selected model is really reliable in real application even without using any labeled target data.
>
> I'm not sure I understand how one would test the selected model on more target domain data. I feel like evaluating mAP on the target datasets, like the authors did, is the best one can do given the limited scope of existing UDA benchmark datasets.

---

> > ### Author Response · Authors · 2024-08-12
> > **Response to Reviewer 4YNt**
> >
> > We thank the reviewer for the constructive comments. We agree with the comments that the target domain labels might be infeasible to obtain in many real-world domain adaptation cases and our method is useful for many real applications. Thank you again, we are happy to receive your comments to further improve the quality of our paper.

---

> > ### Comment · Reviewer_XtC2 · 2024-08-14
> > **"Rating 3: Reject" may be biased**
> >
> > **I agree with Reviewer 4YNt and believe the "Rating 3: Reject " may be biased.** The criteria for "Reject" include "technical flaws, weak evaluation, inadequate reproducibility, and/or incompletely addressed ethical considerations." However, this paper is technically sound, with no apparent flaws. Extensive experiments validate its effectiveness on standard benchmarks, and they provide sufficient reproducibility information without any potential ethical issues. **Therefore, the paper does not meet any of the "Reject" criteria.**
> >
> > Additionally, I am puzzled by the reviewer's concerns about the "uselessness of the DAOD setting," as **there is ample literature supporting the practicality and validity, which has also been recognized by the ML community [1-4]**. I also believe that extending to DG is beyond the scope of this paper and should be considered for the journal-extended version as future work.
> >
> > [1] Learning Domain-Aware Detection Head with Prompt Tuning NeurIPS 2023
> >
> > [2] Learning Domain Adaptive Object Detection with Probabilistic Teacher ICML 2022
> >
> > [3] Decoupled Adaptation for Cross-Domain Object Detection ICLR 2022
> >
> > [4] Synergizing between Self-Training and Adversarial Learning for Domain Adaptive Object Detection NeurIPS 2021

---

> > > ### Author Response · Authors · 2024-08-14
> > >
> > > We sincerely appreciate the comments and the recognition of our work from reviewer XtC2. We are also committed to continuously advancing DAOD towards more practical applications.

---

### Official Review · Reviewer_jxDH · 2024-07-11

**Soundness:** 2
**Presentation:** 3
**Contribution:** 3
**Rating:** 5
**Confidence:** 5

**Summary:**

This paper tackles the problem of model selection in unsupervised domain adaptation for object detection (DAOD). In DAOD, existing methods choose the models (checkpoints) using ground truth data on the target domain, which is impractical in real-world settings. To solve the problem, this paper proposes a model selection method called Detection Adaptation Score (DAS) based on the relationship between flat minima and generalization ability. The proposed DAS consists of two scores: Flatness Index Score (FIS) and Prototypical Distance Ratio (PDR). The FIS calculates the distance between the predictions before and after the perturbation of the model parameters. The PDR calculates the distance between the class prototypes on source and target domains. The experimental results demonstrate the positive correlation between the DAS and mAP. Also, the models chosen by the DAS outperform those chosen by other unsupervised model selection methods.

**Strengths:**

i) This paper is well-written and easy to follow. The motivation for tackling the task is discussed in Sec. 1, and related works are well-summarized in Sec. 2. The ideas and details of the proposed method are clearly described in Sec. 3.

ii) The proposed method is simple yet effective. In addition, it is practically beneficial for not only model selection but also hyper-parameter tuning on DAOD.

iii) The performance of the proposed method is good. The proposed method shows better performance than other unsupervised model selection methods in terms of the mAP of the chosen models. The ablation studies show that each of PDR and FIS contributes to better model selection.

**Weaknesses:**

i) The experiments in this paper show that the proposed method works well on model selection in DAOD. However, I think the proposed method is not limited to DAOD; that is, it can be directly applied to other tasks such as UDA for image classification and semantic segmentation (except IoU calculation in Eq. 2). I would like to see whether the proposed method works well those tasks as well.

I have some concerns about the relationships between the flat minima principal and the proposed method as follows:

ii) The first term of the right-hand side in Eq. (1) is the difference between the risks (i.e., losses) of $h$ and its neighbor $h'$. In contrast, the proposed method calculates the variance in the outputs, not the risks. They are different measures, and the relationships between them are not addressed in this paper although there might be a correlation between them. This paper will be more convincing if mathematical relations between the difference in the risks and the variance in the outputs are derived.

iii) I understand that the PDR evaluates the transferability and discriminability because high PDR means that the prototypes of the same classes are well-aligned between the source and target domains. However, the relationship between better transferability (and discriminability) and flat minima is not clear in this paper. Why do better transferability and discriminability lead to flat minima?

iv) Although it was confirmed that each of FIS and PDR contributes to better model selection in terms of mAP, the evaluation of the correlations between each of FIS and PDR and flatness is missing. Similar to [7], the flatness can be evaluated as the change in risk using GT labels. This paper will be better if it provides the experiments to evaluate the correlations between each of FIS and PDR and flatness. The experiments can address my concerns ii) and iii) indirectly.

**Questions:**

See i), ii), and iii) in Weakness.

i) Does the proposed method work well in other UDA tasks as well?

ii) Is it possible to derive the mathematical relationship between the difference in risks and the variance in outputs?

iii) Why do better transferability and discriminability lead to flat minima?

**Limitations:**

Limitations are adequately addressed in this paper.

---

> ### Author Rebuttal · Authors · 2024-08-07
>
> Thank the Reviewer jxDH for your insightful comments. We appreciate the positive comments regarding the writing, method, and performance. Now we answer the question raised by the reviewer as follows.
>
> **Q1**: Does the proposed method work well in other UDA tasks as well?
>
> **A1** Yes. Our method can also work in other UDA tasks. To verify this, we conduct experiments on semantic segmentation and evaluate the well-known DAFormer method on GTA-5 to Cityscapes. The experimental results in terms of mIoU are shown below. It is shown that our DAS score can choose a better checkpoint of the model with an average mIoU of 64.2% while the last checkpoint of the model only achieves 60.9%. This demonstrates that our DAS method can also work effectively on other UDA tasks.
>
> | Method   | Last | Ours | Imp.$\uparrow$ | Oracle |
> | -------- | ---- | ---- | -------------- | ------ |
> | DAFormer | 60.9 | 64.2 | 3.3            | 65.9   |
>
> **Q2**: Is it possible to derive the mathematical relationship between the difference in risks and the variance in outputs?
>
> **A2**: The variance in outputs serves as an upper bound for the difference in risks. To establish this relationship, let us define the error as $\mathcal{E} _ {h}(o _ {h}, g) = \left| o_ {h}-g \right|$, where $o _ {h}$ is the output of the network $h$ and $g$ is the ground truth. Similarly, error of network $h'$ is denoted as $\mathcal{E} _ {h'}(o _ {h'}, g) = \left| o_ {h'}-g \right|$. By applying the triangle inequality, we derive:
> $$
> \left|\mathcal{E} _ {h}(o_{h}, g) - \mathcal{E} _ {h'}(o _ {h'}, g) \right| = \left|\left| o_{h} - g \right| + \left| o _ {h'} - g \right| \right|  \leq \left| o _ {h} - o _ {h'} \right|
> $$
> Consequently, minimizing the variance in outputs effectively constrains the difference in risks, providing a mechanism to bound the variability in predictions across networks.
>
> We appreciate your insightful comments and will delve further into these mathematical underpinnings in our revised manuscript to illustrate the implications of variance in outputs on risk difference.
>
> **Q3**：Why do better transferability and discriminability lead to flat minima?
>
> **A3**: This is a good question. I would like to gently point out that variance in outputs primarily indicates **flatness**. If the labels are available, we can use the labels to calculate the error on the target domain so as to ensure the property of **minima**. Without using target labels, we estimate the generalization error on the target domain through a domain adaptation theory, where the target error is bounded by source error, domain distance, and a constant term. The transferability and discriminability are widely used to improve the performance on the target domain. With the both properties of flatness and minima on the target domain, we can obtain the flat minima without using target labels. Thanks for your valuable feedback, we will add more explanations to make it clearer in the final paper.
>
> **Q4**: The correlations between each of FIS and PDR and flatness with GT labels.
>
> **A4**: Thank the reviewer for pointing this out. We provide an experiment to verify the positive correlation between our FIS and PDR and flatness with GT labels. In particular, we show the correlation coefficient between them on AT weather adaptation. It shows that the proposed FIS is well correlated with the GT flatness with a coefficient of 0.64, demonstrating that our DAS score can be an appropriate estimation for flatness with GT labels. However, PDR represents the target error by assessing the transferability and discriminability thus it doesn't well correlate with the flatness of GT labels, as the coefficient is 0.45. It is by combining FIS and PDR that we could find the flat minima without accessing target labels while achieving accurate model selection for DAOD models.

---

> ### Author Response · Authors · 2024-08-13
>
> Dear Reviewer jxDH,
>
> We hope this message finds you well. As the discussion period is ending soon, we would like to bring to your attention that we have submitted our response to your questions. We carefully consider your valuable feedback and suggestions, and hope the response well addressed your concerns.
>
> We are truly thankful for the comments on our work.
> If you find our responses to be satisfactory, we would greatly appreciate it if you could take this into account when considering the final score.
>
> Best regards,
>
> Authors of Submission 7161

---

> > ### Comment · Area_Chair_CtVr · 2024-08-13
> >
> > Reviewer jxDH, do you have any additional questions or feedback?

---

> > > ### Comment · Reviewer_jxDH · 2024-08-13
> > >
> > > I would like to thank the authors for their careful rebuttal. Because my concerns were well addressed, I'm raising my rating to Borderline Accept.

---

> > > > ### Author Response · Authors · 2024-08-14
> > > >
> > > > We greatly appreciate your response and the recognition of our work. We will refine our work in accordance with your recommendations to further improve the final version.

---

### Official Review · Reviewer_4YNt · 2024-07-11

**Soundness:** 3
**Presentation:** 3
**Contribution:** 3
**Rating:** 8
**Confidence:** 4

**Summary:**

The paper introduces a new metric ("DAS") for unsupervised model selection in domain-adaptive object detection. DAS consists of two components: a flatness index scores that approximates the flatness of the loss landscape in the target domain by measuring prediction agreement across perturbed model parameters, and a prototypical distance ratio that measures the ratio of inter-class and intra-class feature dissimilarity. DAS is evaluated for model selection and hyperparameter optimization and correlates well with true model performance.

**Strengths:**

The paper presents a significant step forward in the underexplored area of unsupervised model selection in domain adaptive object detection. This is evidenced by considerable gains in object detection performance on the target domain, compared to baseline approaches. Although concepts similar to the two main contributions (flatness index score & prototypical distance ratio) have previously been in explored in the literature, their combination and application to unsupervised model selection in domain adaptive object detection is novel. The overall structure of the paper is really clear, and the evaluation is extensive and informative.

**Weaknesses:**

The writing quality should be improved (e.g. general grammar, typos such as "Faster RCCN", sometimes inconsistent terminology such as "Prototype Distance Ratio" / Prototypical Distance Ratio"). Apart from writing quality, my only significant concern is the apparent invariance of DAS performance towards lambda, which raises the question of whether the flatness index score and prototypical distance ratio measure different aspects in practice and the degree to which they are complementary.

**Questions:**

How is PDR or d_inter defined for single-class benchmarks such as Sim10k?

DAS seems to work consistently very well for AT, while performance with other frameworks is less consistent (very apparent in Figure 4). Do you know why this might be the case?

It might make sense to include the oracle checkpoint in the main evaluation, especially tables 1-3. This would make it easier to see how much of the gap to the oracle is closed.

Please clarify what is meant by "ES" in table 2.

It would be interesting to see analogues for Figures 2 and 5 for Cityscapes and Sim10k in the appendix.

**Limitations:**

Limitations and societal impact are adequately discussed in the appendix.

---

> ### Author Rebuttal · Authors · 2024-08-07
>
> Thank Reviewer 4YNt for constructive comments and valuable feedback. We appreciate the recognition of our work including the support for the novelty, structure, and evaluation. Now, we address the reviewer's concerns as follows.
>
> **Q1**: The writing quality should be improved.
>
> **A1**:We sincerely thank your constructive comments. We have corrected the grammar issues, typos, and inconsistent terminology. We also carefully proofread the paper.
>
> **Q2**: Apart from writing quality, my only significant concern is the apparent invariance of DAS performance towards lambda, which raises the question of whether the flatness index score and prototypical distance ratio measure different aspects in practice and the degree to which they are complementary.
>
> **A2**: From the generalization bound theory for flat minima present in the paper, the FIS and PDR measure the flatness and the minimum, thus they are theoretically evaluating the models in two aspects. In Table 5, the selected model mAP doesn't vary too much with different $\lambda$ setting. This might have contributed to when the model convergences to some degree, the two terms FIS and PDR represent have been both optimized well. Thus they both show relatively stable results on the mAP. This is why we use PCC between DAS and the model to measure the subtle differences. On the other hand, the PCC has obvious changes with the variation of $\lambda$. This indicates the $\lambda$ indeed changes the DAS and influences the correlation between the proposed DAS and ground truth performance. To examine the complementary, we have conducted an ablation study in Table 4 from the main paper. It is shown that FIS only achieves $0.48$ PCC. With the help of PDR, our DAS improved the PCC from $0.48$ to $0.67$, demonstrating the complementary between FIS and PDR.
>
> **Q3**: How is PDR or d_inter defined for single-class benchmarks such as Sim10k?
>
> **A3**: Although Sim10k only has a single class, the object detection model also considers the background class, thus the PDR or d_inter can be calculated. We will add more descriptions in the final paper to reduce the confusion.
>
> **Q4**: DAS seems to work consistently very well for AT, while performance with other frameworks is less consistent (very apparent in Figure 4). Do you know why this might be the case?
>
> **A4**: The main reason lies in that the performance gap of other frameworks between the last checkpoint and Oracle results is narrow, so it gives our method a relatively small improvement space. For example, the average performance gap of AT over three settings is 8.68%, while that of MT over three settings is 1.28%. In particular, a narrow performance gap gives less tolerance for potential uncertainty of the DAS estimation. Therefore, the improvements with other frameworks show less consistent improvement like AT. Nevertheless, the proposed method can still select better checkpoints in most cases and checkpoints that are at least comparable with last checkpoints in very few cases.
>
>
> **Q5**: It might make sense to include the Oracle checkpoint in the main evaluation, especially tables 1-3. This would make it easier to see how much of the gap to the oracle is closed.
>
> **A5**: We thank your constructive comments. We will add the orcale checkpoint in tables 1-3. We present Tables A1 and A2 in the attached PDF in the rebuttal to show the added version of Tables 1 and 2 in the main manuscript.
>
> **Q6**: Please clarify what is meant by "ES" in table 2.
>
> **A6**: ES[40] indicates the entropy score, i.e., using the entropy of the prediction from the classification branch to select the checkpoints. We have clarified this in the experiment section.
>
> **Q7**: It would be interesting to see analogues for Figures 2 and 5 for Cityscapes and Sim10k in the appendix.
>
> **A7**: We provide more examples on Sim10k to Cityscapes and Cityscapes to Foggy Citysacpes adaptation in Figures A1 and A2 in the attached PDF. Similar to Figure 2 in the main paper, the proposed DAS score correlates well with the performance of DAOD checkpoints compared with other baselines.

---

> > ### Comment · Reviewer_4YNt · 2024-08-13
> >
> > Based on the author's responses to my questions and the additional information provided in the rebuttal, I'm raising my score to a strong accept.

---

> > > ### Author Response · Authors · 2024-08-14
> > >
> > > Thank you for your timely and valuable feedback. We are grateful for your recognition of our work. We will carefully revise the final version based on your valuable feedback.

---

### Official Review · Reviewer_ood2 · 2024-07-12

**Soundness:** 3
**Presentation:** 3
**Contribution:** 3
**Rating:** 5
**Confidence:** 4

**Summary:**

This paper delves into an unsupervised evaluation problem in Domain Adaptation Object Detection (DAOD). To solve this problem, this paper proposes a method based on the flat minima principle, named Detection Adaptation Score (DAS), which can measure the flat minima without using target labels. Specifically, the proposed method is composed of a flatness index score (FIS) and a prototypical distance ratio (PDR) to assess the flatness and measure the transferability and discriminability of the models. Experiments validate the effectiveness of the proposed method.

**Strengths:**

-	This paper is well motivated. And this paper is the first work to evaluate DAOD without using target labels.
-	The proposed method is effective and reasonable.

**Weaknesses:**

-	In line 49, the authors should explain why these methods fail to evaluate the object detection model.
-	In Eq.2, since KL term and IoU term are two metrics to evaluate the matching costs of two different tasks, object classification and box regression, why there is no a balance coefficient, such as d=KL-\lambda IoU. If \lambda=1 is the best?
-	The proposed FIS is easy coming to mind, which uses parameter perturbation to evaluate flatness minima without using target labels. And the proposed PDR to use prototype-based domain alignment is also used as a training method in the existing prototype-based domain alignment for DAOD, such as [1].
-	The proposed FIS and PDR are used for unsupervised evaluation. However, from another perspective, these methods can also be used as unsupervised training methods. If these kinds of methods are used for unsupervised training, can they be used as unsupervised evaluation methods as the same time? For example, if the benchmark DAOD framework is a prototype-based domain alignment method, can PDR, a prototype-based domain alignment evaluation method, serves as an unsupervised evaluation? As I can see in line 261, this paper only uses adversarial training or self-training methods for unsupervised domain adaptive training? More different kinds of benchmark DAOD framework are necessary to evaluate the scope of the proposed evaluation methods.

[1] Cross-domain detection via graph-induced prototype alignment. CVPR 2020.

**Questions:**

See the weaknesses, especially the forth question.

**Limitations:**

Limitations had been discussed.

---

> ### Author Rebuttal · Authors · 2024-08-07
>
> We thank Reviewer ood2 for the valuable feedback and insightful comments. We appreciate the reviewer's positive comments regarding motivation and method. We now clarify the reviewer's concerns as follows.
>
> **Q1**: Explain why these methods fail to evaluate the object detection model in line49.
>
> **A1**: The methods referenced in line 49 rely on classifier-specific properties like predicted confidence and entropy, which are tailored for classification tasks. In object detection, however, the evaluation involves not only classification but also the precise localization of objects within an image. This crucial distinction renders these methods ineffective in fully assessing an OD model. By incorporating your suggestion into our paper, we will explicitly clarify this difference between classification-centric evaluation methods and the multifaceted demands of OD evaluation.
>
> **Q2**: In Eq.2, since KL term and IoU term are two metrics to evaluate the matching costs of two different tasks, object classification and box regression, why there is no a balance coefficient, such as d=KL-\lambda IoU. If \lambda=1 is the best?
>
> **A2**: Thank you for regarding the potential absence of a balance coefficient in Eq. (2) to weigh the KL divergence and IoU metrics. We do not deliberately add a balance coefficient following the previous works (e.g., DETR) to combine the classification and localization cost. As suggested by the reviewer, we conducted experiments on AT weather adaptation to assess the impact of tuning the balance coefficient. The results are shown in the table below. Our findings suggest that the parameter adjustment can actually influence performance, while setting the balance coefficient to 1 is a fairly good choice, indicating a balanced consideration of classification and localization within our framework.
>
> | $\lambda$ |0|0.1|1|10|100|
> |-|-|-|-|-|-|
> | mAP |48.2|49.3|49.3|48.7|48.5|
>
> **Q3**: The proposed FIS is easy coming to mind, which uses parameter perturbation to evaluate flatness minima without using target labels. And the proposed PDR to use prototype-based domain alignment is also used as a training method in the existing prototype-based domain alignment for DAOD, such as [1].
>
> **A3**: We would like to highlight our paper is essentially different from previous works. We address the unsupervised model selection (UMS) problem for DAOD. Previous works [1] design effective methods for addressing the domain gap for object detection, while we address the UMS problem for DAOD through a flat minima principle, i.e., models that locate the flat minima region in the parameter space usually exhibit excellent generalization ability. Different from the traditional flat minima methods that require labeled data to evaluate how well a model is located at the flat minima region, we show via the generalization bound that the flatness can be deemed as model variance, while the minima depends on the domain distribution distance for the DAOD task. To this end, although some strategies in our paper may seem to be similar to existing works, the task and motivation are distinct from the existing works.
>
> **Q4**: If these kinds of methods are used for unsupervised training, can they be used as unsupervised evaluation methods as the same time? For example, if the benchmark DAOD framework is a prototype-based domain alignment method, can PDR, a prototype-based domain alignment evaluation method, serves as an unsupervised evaluation?
>
> **A4**: To effectively evaluate the DAOD models, we designed a DAS including FIS and PDR, which is derived from a generalization bounding for flat minima. Thus, they evaluate the DAOD models in different aspects. Additionally, existing work [1] use the prototype-based alignment method to minimize the domain gap between domains. The prototype estimation in existing work only utilizes the samples in a mini-batch during the model training. In contrast, we use the entire dataset to estimate the prototypes, which is more robust and has better generalization ability. To verify this, we conduct experiment results on Sim10k to Cityscapes (S2C) for [1]. The experimental results are presented below. It is shown that the proposed method still works when DAOD frameworks also optimize the prototype-based distance, i.e., choosing a checkpoint with 44.8% mAP (our PDR) while the last checkpoint is 43.1% mAP. Moreover, our DAS with FIS and PDR can further improve the performance, demonstrating the effectiveness of the FIS. The FIS is proposed by us in this paper, and we have not found any DAOD methods using it as a training method.
>
> |Method|Setting|Last|PDR|DAS|Oracle|
> |-|-|-|-|-|-|
> |GPA|S2C|43.1|44.8(+1.7)|45.8(+2.7)|47.0|
>
>
> **Q5**: As I can see in line 261, this paper only uses adversarial training or self-training methods for unsupervised domain adaptive training?
>
> **A5**: We evaluate DAOD methods with adversarial training or self-training because they are classical paradigms and achieve promising results in DAOD scenarios. To demonstrate the generalization of the proposed DAS method to other DAOD frameworks, we evaluate our method on GPA[1] and SIGMA++[2] which minimize the domain gap by prototype-based domain alignment and graph matching, respectively. The results are shown in the below table, we can see that our method is able to select better checkpoints for those two DAOD methods (2.7% and 2.2% improvements, respectively) beyond adversarial learning and self-training paradigms.
>
> |Method|Setting|Last|Ours|Imp.$\uparrow$|Oracle|
> |-|-|-|-|-|-|
> |GPA|S2C|43.1|45.8|2.7|47.0|
> |SIGMA++|C2F|39.5|41.7|2.2|41.7|
>
> * We attempted but could not reproduce the GPA results on C2F using the released code. This problem is found by other researchers in the issue of the repo. Thus we choose the setting of S2C.
>
> [1] Cross-domain detection via graph-induced prototype alignment. CVPR 2020.
>
> [2] SIGMA++: Improved semantic-complete graph matching for domain adaptive object detection. TPAMI 2023.

---

> > ### Comment · Reviewer_ood2 · 2024-08-13
> >
> > Thanks for the rebuttals from the authors, which address my concerns to some extents. I tend to keep my original score, which leans to positive already.

---

> > > ### Author Response · Authors · 2024-08-14
> > >
> > > Thank you for your prompt response and your recognition of our work. We greatly appreciate your time and effort in reviewing our submission. We will carefully incorporate your feedback to further enhance the quality of our final work.

---

> ### Author Response · Authors · 2024-08-13
>
> Dear Reviewer ood2,
>
> We hope this message finds you well. As the discussion period is ending soon, we would like to bring to your attention that we have submitted our response to your questions. We carefully consider your valuable feedback and suggestions, and hope the response well addressed your concerns.
>
> We are truly thankful for the comments on our work.
> If you find our responses to be satisfactory, we would greatly appreciate it if you could take this into account when considering the final score.
>
> Best regards,
>
> Authors of Submission 7161

---

### Author Rebuttal · Authors · 2024-08-07

Dear reviewers,

We would like to thank all the reviewers for their insightful comments and constructive feedback which have significantly enhanced the quality of our work. We appreciate that the reviewer acknowledges the advantages of our work: "**This paper is well motivated. The proposed method is effective and reasonable**"(Reviewer ood2), "**The paper presents a significant step forward in the underexplored area of unsupervised model selection in domain adaptive object detection**"(Reviewer 4YNt), "**The proposed method is simple yet effective**"(Reviewer jxDH), "**The proposed strategy is based flat minima which is therefore rational.**" (Reviewer 9Lzv), "**The perspective about model selection for DAOD is interesting**"(Reviewer XtC2).

On the other hand, we have diligently addressed all the concerns raised by the reviewers. In the attached PDF file, we also provide the tables with Orale results for easier to see how much of the gap to the Oracle is closed and more comparison of different unsupervised model evaluation methods for DAOD.

Best Regards,

Authors of Submission 7161.

---

### Decision · Program_Chairs · 2024-09-25

**Decision:**

Accept (poster)

**Comment:**

Though the reviews are mixed, the majority of the reviewers recommend acceptance and emphasize the importance of the topic (unsupervised model selection for domain adaptation), which has been previously under-explored in the literature. The authors propose a novel method for addressing this which performs close to optimal when assuming access to labels in the target domain. The AC team appreciates the value of the contribution and thus agrees with the majority of the reviewers that the paper should be accepted.